# Deceptive Seduction by *Femme Fatale* Fireflies and Its Avoidance by Males of a Synchronous Firefly Species (Coleoptera: Lampyridae)

**DOI:** 10.3390/insects15010078

**Published:** 2024-01-22

**Authors:** Yara Maquitico, Jazmín Coronado, Andrea Luna, Aldair Vergara, Carlos Cordero

**Affiliations:** 1Posgrado en Ciencias Biológicas, Universidad Nacional Autónoma de México, Ciudad Universitaria CDMX 04510, Mexico; yaramaquitico@gmail.com; 2Licenciatura en Biología, Universidad Autónoma Metropolitana-Unidad Xochimilco, Ciudad Universitaria CDMX 04960, Mexico; jazmincoronado0@gmail.com; 3Licenciatura de Biología, Facultad de Ciencias, Universidad Nacional Autónoma de México, Ciudad Universitaria CDMX 04510, Mexico; andrealunam@ciencias.unam.mx; 4Facultad de Estudios Superiores Iztacala, Universidad Nacional Autónoma de México, Tlalnepantla EDOMEX 54090, Mexico; doctorlobo18@gmail.com; 5Instituto de Ecología, Universidad Nacional Autónoma de México, Ciudad Universitaria CDMX 04510, Mexico

**Keywords:** aggressive mimicry, deception, hunting success, predation avoidance, sexual signaling

## Abstract

**Simple Summary:**

Predatory female fireflies of several species in the genus *Photuris* attract males from other firefly species by responding to their flash signals, and then, the females try to capture and feed on the attracted males. Predatory female *Photuris*, called “femmes fatales”, are considered a serious threat to males of other firefly species; however, there are very few quantitative data on the strength of this menace. We measured the attraction of *Photinus palaciosi* males to predatory females of *Photuris lugubris* in the field, as well as the number of prey captured. We observed that females, in general, attract several males of the prey species but capture relatively few; only nine out of 92 (9.8%) observations of predatory females resulted in successful capture. Our observations show that the low hunting success of *Photuris* females could be explained in part by male behaviors. First, the males attracted behave in a way that we call “cautious” or “hesitant”, which could prevent attacks, and second, these males also behave in ways that probably reduce the success of attacks at close distance, such as approaching in flight and “dropping” in the face of an imminent attack. Female *Photuris* also behave in ways that probably improve their success in attracting and capturing prey, such as perching in or near display areas of their prey species, and partially concealing their own lantern on the substrate to avoid revealing their much larger size and/or to better mimic the responses of *Photinus* females. Our observations are thus consistent with the idea that predators and prey are engaged in a coevolutionary race of adaptation and counter-adaptation.

**Abstract:**

*Photuris* female fireflies attract males of different firefly species by responding to their flashing signals; then, they try to capture and feed on them. This aggressive mimicry is considered a major selective pressure on the communication systems of the fireflies of the American continent. The intensity of this selective pressure is a function of its efficiency in prey capture. In this study, the rates of attraction and capture of males of the synchronous firefly *Photinus palaciosi* by the predatory females of *Photuris lugubris* are reported. Although the females attract numerous males, their hunting success is low. This result is consistent with the few previous measurements published. In agreement with the predicted coevolutionary race between predator and prey, behaviors consistent with predation avoidance in *P. palaciosi* and increasing prey encounters and prey deception by *P. lugubris* were observed.

## 1. Introduction

Deception is a ubiquitous feature of life [1,2]; evolutionarily speaking, deception has been selected to improve escape from natural enemies [1,3] and increase reproduction (as in pollination by deceit [1]) or food acquisition [1]. A fascinating example of interspecific deception is aggressive mimicry, which occurs when “the mimic signals a fitness benefit to the receiver and the mimic’s signal is deceptive” ([4], p. 4). Aggressive mimicry as an aid to capture prey has been described in a wide range of animals, including spiders [5], aphids [6], katydids [7], fish [8,9], and *Homo sapiens* [10]. In North America, predatory firefly females from species belonging to three genera of the subfamily Photurinae (*Bicellonycha*, *Crematogaster*, and *Photuris*) emit bioluminescent responses to the flashing signals of males of other firefly species (several species of *Photinus* are common prey, although males from other genera, including other *Photuris* species, are also hunted); in this way these “femmes fatales” attract the males and then try to capture them and feed on them [11,12]. By feeding on the males of other species, predatory females not only obtain nutrients and energy but also defensive steroid pyrones called lucibufagins (molecules that *Photuris* cannot produce) that protect the female [13] and her eggs [14] from predators. Thus, firefly femmes fatales [11,12,15] are classical and extreme examples of aggressive mimicry: classical because the femme fatale sends fake signals indicating the presence of a sexually receptive female and extreme because it can result in the deceived individual being killed.

In many cases, *Photuris* females mimic the specific response signals of the females of the prey species [12,15], and individual females of some species have been shown to mimic the specific response of females from several species depending on the identity of the potential prey [12,15,16]. These studies led the most important researcher of Photurinae and femme fatale behavior, the late Prof. James Lloyd, to think that femmes fatales are the source of selective pressures that have been crucial for the evolutionary divergence of bioluminescent signals in prey firefly species of the American continent ([15]; see also [12,15,17,18]). Females of predatory *Photuris* have morphological adaptations apparently designed to subdue and consume their prey [19], and recent molecular studies have identified genes involved in resistance to lucibufagins [20] and several candidate genes that could represent diverse physiological adaptations for the consumption of potentially toxic prey [21]. Thus, it is expected that *Photuris* predation is the source of selective pressures promoting the evolution of countermeasures in their prey fireflies [15,18], which resulted in a coevolutionary process that could be responsible for the complexity and plasticity frequently observed in the communication systems of fireflies [12,16,18,22,23,24,25,26]. Since, in many places, several potential prey firefly species coexist, sometimes with more than one species of femmes fatales, the coevolutionary process, rather than a product of pairwise interactions, is more likely to be of the type known as “diffuse coevolution” [27,28]. In this context, some prey counter-adaptations could select for an increased degree of deceptiveness in the signals produced by *Photuris* females. This has led to the study of the match between the response signals produced by females of prey species and the deceptive responses emitted by femmes fatales, implicitly considering the degree of match as a measure of the degree of deceptiveness (see review in [12]). However, a perfect copy of the signal is not necessary for successful mimicry; instead, only the components used by the receiver to assess the signal need to be mimicked [4].

On the other hand, since aggressive mimicry selects for male prey counterstrategies to avoid being attracted and captured, a more direct way of measuring the efficiency of the deceptive signals of *Photuris* females is measuring their success in attracting, capturing, and consuming prey. These measurements are more directly related to the selective pressures exerted and the adaptations developed by both predator and prey. When the courting areas of femmes fatales and prey fireflies do not overlap, it is reasonable to expect that hunting females move to places where the probability of finding prey is higher and then start to attract and attempt to capture prey. Previous studies (summarized in [12]) and a recent study of *Photuris lugubris* [29] indicate that mated females move from their own courtship display areas to, or close to, the courtship display areas of their prey fireflies. However, the consequences of this movement for prey encounters, although apparently obvious, have not been quantified. Similarly, rates of prey attraction and capture by firefly femmes fatales are rarely reported. Lloyd [16] studied *Photuris versicolor* hunting two species of *Photinus* and two species of *Photuris* in the field and observed wide interspecific variation in the rates of capture of the different prey species (4%, 9%, 20%, and 54%). However, Lloyd did not report or discuss the causes of this variation. In a laboratory experiment, Lewis et al. [30] observed large differences in the consumption rates of females of several *Photuris* species confined with males from eight species of Lampyridae, including five species of *Photinus* (one of them, *P. carolinus*, a synchronous firefly). The causes of this variation were not determined, but the fact that the species with low consumption rates showed evidence of having been attacked suggests between-prey differences in defensive chemistry, nutritional value, or ability to escape once captured. Furthermore, since only 11 females from at least three *Photuris* species were studied, interspecific differences in hunting ability cannot be ruled out. Finally, Lloyd [12] also reported an observation of one female *Photuris carrorum* that answered the signals of twenty-five male *Photinus macdermotti* before capturing one (a success rate of 3.8%), although he did not mention if the twenty-five males that were not captured approached the female or if they were unsuccessfully attacked.

In this paper, we report field measurements of rates of attraction and capture for male *Photinus palaciosi*, a synchronous firefly from the mountains of Central Mexico [31], by *Photuris lugubris* femmes fatales [29]. This study is part of a broader project aimed at understanding the evolution of the femme fatale behavior of Photurinae fireflies.

## 2. Materials and Methods

### 2.1. Firefly Natural History in “Rancho del Valle”

This study was conducted in the pine–oak forest of “Rancho del Valle”, a private ecotourism ranch located in the village of Santiago Cuauhtenco, Municipality of Amecameca, Estado de México, Mexico. This section is based on observations made during three mating seasons (2021 to 2023). The mating season of *P. lugubris* (*Photuris* hereafter) starts about the middle of May, when males begin flying and signaling (flashing); one or two weeks after, the first females are observed perching in vegetation at heights ≤ 2 m exchanging signals with conspecific males (i.e., the species is protandrous). During this period, the fireflies occupy a relatively small section of the ranch (<1 ha) dominated by grasses and planted pine trees of heights ≤ 2 m; larger trees are virtually absent in this area [29]. Both males and females can fly, but females remain perched during the nightly courtship period. Most of the courtship and mating activity occurs between 19:30 and 22:00 h (although a few individuals have been observed as early as 19:00 h and later than 22:00 h). To the human eye, males and females are very similar and can be distinguished only by their behavior and the type of bioluminescent signals they produce: males produce intermittent flashes while flying, and the perching females respond with glows produced at much lower rates. When a male succeeds in approaching a female, he briefly touches her with his front legs and antennae and quickly tries to mount her and copulate. The mating season of this species ends about the second week of June; after this time, females move to areas where male *Photinus* are common and start hunting them using aggressive mimicry [29], while male *Photuris* are only occasionally observed courting hunting females. However, in 2023, the rainy season started late in Mexico, adult emergence dates were somewhat delayed, and more males were observed courting hunting females. The first author had the impression that these males were more “cautious” when courting hunting females, and the signaling interactions lasted much longer. Previous studies led us to think that the hunting females were already mated [32,33]. However, females mate multiple times [29], and thus, it is possible that some hunting females are still sexually receptive. In support of this hypothesis, five males were observed mating with hunting females, and in one case, the female discarded the partially eaten prey to mate (this was the first female observed capturing a male in 2023). In contrast to the mating period, during the hunting period, female *Photuris* are observed until 23:00 h. The last *Photuris* females can be observed between the second and third week of July.

*P. palaciosi* (*Photinus* hereafter), a synchronous firefly, is a protandrous species (females appear around a week after the first males) whose females cannot fly because they have reduced wings (i.e., they are brachypterous) [31]. The mating season of this species starts between the first and second week of June, so males appear approximately when *Photuris* females start hunting. Courtship and mating occur more commonly in areas covered by the foliage of trees, all over the forest. Males usually fly at heights ≤ 2 m, flashing intermittently in search of females perching in low vegetation (<0.5 m). Females respond with glows produced at much lower rates. When a male succeeds in exchanging signals with a female, he flies toward the female, lands close to her, and finally approaches walking. Once he contacts the female, he starts tapping the dorsum of her body with his legs and antennae in a way suggesting contact courtship; this interaction can extend for several minutes until copulation ensues or until the male departs, possibly because the female rejects copulation (females must cooperate by opening her valves to allow intromission of the aedeagus). Although a few males can be observed flying as early as 19:00 h, most of the courtship activity occurs between 19:30 and 22:00 h, with a few signaling individuals observed later. The mating season of *Photinus* finishes about the first week of August.

The two focal species of this study coexist with a third firefly, *Photinus extensus*. This species is less abundant than the others, it is possibly protandrous, and its females are also brachypterous. Males of this species begin to be observed by the middle of June, and its mating season finishes by the end of July. The nightly courtship period is shorter than that of the other two sympatric species, between 19:00 and 21:00 h. The flying males emit intermittent flashes that are responded to with glows by the females perched on herbs at heights ≤ 0.5 m. The courtship and mating of this species were observed all over the forest. In three consecutive mating seasons (2021–2023), *Photuris* females have never been observed attacking or feeding on male *P. extensus*; in fact, no male *P. extensus* has been observed flying toward or inspecting a female *Photuris*. *P. extensus* males are larger than *Photuris*, which, in turn, is larger than *P. palaciosi*. The first author once exposed a male *P. extensus* to a female *Photuris* (in a container similar to those previously used to test that *Photuris* females feed on *Photinus* males [29]) and observed that the female approached the male only once without attacking, and the male was alive and intact the next morning (in contrast, 15 out of 16 *Photuris* females attacked the *P. palaciosi* males they were confined with).

### 2.2. Field Study

From June 30 to July 22, 2023, between 19:00 and 23:00 h (an interval that covers the nightly display period of *P. palaciosi* and *P. lugubris*), two to four trained observers walked along paths in the forest looking for *Photuris* females. The paths were those where fireflies have been observed in previous years [29] and are usually located in or near *Photinus* courtship sites (personal observation). Once a female *Photuris* was found, usually by detecting one of her glows, one observer continuously monitored the behavior and interactions of the female (thus, we used the focal sampling method [34]) from a distance between 50 cm and 1 m until the female either flew away from the site or stopped performing relevant activity (i.e., responding to male flashes), which usually occurred after the nightly courtship display period of *Photinus*. Since the population of *Photuris* is small (personal observation), and it is the only one that has been found so far coexisting with *Photinus palaciosi*, we decided not to mark *Photuris* females to prevent potential negative effects on the fireflies. Furthermore, hunting females are easily disturbed and fly away, drop to the ground and hide, or stop producing signals when touched. The observations were recorded with the audio-recorder app of the observer’s cell phone; occasional photographs and short videos were also taken.

The number of *Photinus* males that flew within 3 m around the female was recorded. These males were usually detected via their flashes; this distance was used because Lloyd ([11], p. 654) mentions that female fireflies rarely answer to flashing males at distances greater than 3 m. These numbers allowed us to measure the effect of local prey density on the attraction and capture of prey and, thus, indirectly, the effect of female location on hunting success. The following behaviors were recorded: (1) *Communication*: An event of communication was considered to occur when individual A (*Photinus* flying male or *Photuris* perched female) responded with a flash or glow to a signal emitted by individual B (*Photuris* perched female or *Photinus* flying male), with B then responding with a glow or flash to the signal emitted by A; the pair could continue exchanging signals as a part of the same communication event. A female *Photuris* was considered to answer a specific *Photinus* male when she responded with a glow to one of his flashes and moved her lantern in his direction (this behavior is like that observed during courtship, both in *Photuris* and *Photinus*). If the female had more than one male *Photinus* flashing around her, the female was considered to answer if she produced glows in response to some of the flashes; in these cases, she usually directs her lantern to a (preferred?) male. A male *Photinus* was considered to answer a female *Photuris* when he produced flashes after the female glowed and remained close to her (<1 m). (2) *Approach*: A male was considered to approach a female when he changed the trajectory of his flight in the direction of the signaling female during or after a communication event; the male could approach the female flying or, at some point, alight on the ground or on a plant and approach walking. It was recorded if the male approached the *Photuris* female at 5 cm or less because these are the more frequent distances at which we have observed females initiating an attack. (3) *Attack*: The female suddenly approaches a nearby male (usually when the distance between them is ≤5 cm), either walking or jumping, and tries to capture him. Although the attacked males usually exchanged signals with the female before the attack, the female turned off her lantern sometime before the attack. It was recorded if the male escaped the attack or if he was captured and eaten by the female. Female *Photuris* eat most of the tissues of their prey, discarding only the exoskeleton; eating one prey takes several hours.

### 2.3. Statistical Analyses

The data were analyzed considering that hunting is a sequential process in which females first locate themselves in areas of higher probability of finding prey and then attract males and finally attempt to capture them. The potential relationships between the following variables were calculated with Pearson correlations: (1) the number of males passing at a distance ≤ 3 m from a *Photuris* female and the number of males that approached that female to a distance ≤ 5 cm; (2) the number of communication events involving a female and the number of males that approached that female to a distance ≤ 5 cm. These correlations measured the potential effect of local prey density (and thus, indirectly, female location) and the aggressive mimicry of signals on the attraction of prey within attack distance. Fisher’s exact tests were used to investigate the potential relationship between (3) the number of males that approached a female at a distance ≤ 5 cm and the probability of a successful attack and between (4) the number of attacks performed by a female and the probability of a successful attack. All tests were performed using the open-access statistical software available on the website www.socscistatistics.com, accessed on 2 October 2023.

## 3. Results

A total of 92 observations of *Photuris* females were made. Since the females were not marked, it is not known how many different females were studied. Some females were observed in, or very close to, places where females were observed on previous nights, and for this reason, we suspect that at least some females were observed on more than one night. The total time looking for females was 153.2 h, and the amount of time observing *Photuris* females was 56.8 h, for a total of 210 h of fieldwork.

Prey density, measured as the number of male *Photinus* observed flying at 3 m or less from the female *Photuris*, had a positive effect on the number of males that approached the female close enough to risk being attacked (≤5 cm) (Figure 1; *r* = 0.46, *p* < 0.00005, *n* = 71; sample size is less than 92 because one of the observers did not record the number of males passing at a distance ≤ 3 m). The number of communication events, a direct measure of aggressive mimicry, also had a positive effect on the number of males that approached a female *Photuris* at attack distance (Figure 2; *r* = 0.65, *p* < 0.00001, *n* = 71). Although in some cases males approached females without a previous communication event (for example, sometimes they appeared to be attracted by the flashes of other males close to the female), in 83 of the 92 observations (90.2%), there was at least one communication event between a female *Photuris* and a male *Photinus*.

Only 9 of the 92 observations (9.8%) ended in successful hunting. The number of successful attacks was independent of the number of *Photinus* males that approached a female *Photuris* at an attack distance (≤5 cm) (Figure 3; Fisher’s exact probability test, *p* = 1). Although females were observed attacking males up to six times in a night, all nine successful hunts occurred in the first attack of the night (Figure 4; Fisher’s exact probability test, *p* = 0.041).

The behavior frequently observed during *Photuris*–*Photinus* interactions helps us understand the low rates of hunting success. *Photinus* males attracted by *Photuris* females behave as if they are “hesitant” (or very “cautious”) to approach *Photuris* females because they frequently remain in flight exchanging signals for several minutes at a relatively close range (<1 m) but without further approaching. This behavior is different from the direct male approaches observed when *Photinus* males court females of their own species (personal observations). Also, males frequently, upon alighting, approach the females slowly, in contrast to the faster approximations observed in sexual interactions between male and female *Photinus* (personal observations). Generally, when a female *Photuris* approaches a *Photinus* male that is flying or perching close to her, he suddenly “drops” several centimeters, possibly to avoid contact; in these cases, the female does not follow the male. On the other hand, hunting females appear to partially conceal their lanterns by pushing them against the surfaces of their perches (usually leaves), which makes sense considering that female *Photuris* are much larger than *Photinus* females, although this behavior could also help them to mimic the response pattern. Additionally, sometime before an attack, females usually turn off their lanterns. This behavior led one of the reviewers of this paper to the reasonable suggestion that the hunting behavior of *Photuris* is best described as a combination of aggressive mimicry and, in the instants before the attack, stalking.

Besides the nine females observed hunting and feeding on their prey, four other *Photuris* females were found already feeding on a male *Photinus*. Most of the observations of successful hunting occurred during the first 10 days of observation (10/13; Figure 5), approximately corresponding to the first half of the sampling period. Since there were fewer observers per night in the second half of the study season (Figure 5), this number could be an underestimate, although the observers had the impression that the number of predatory females was decreasing during the second half of the study period. Most of the successful attacks (8/9) were observed before 21:00 h (Figure 5), which is consistent with the observation that successful attacks were the first attacks of the night (Figure 4).

## 4. Discussion

Since the courting and mating area of *Photuris* in the study site is very small and barely overlaps with the courting area of *Photinus*, hunting females move to, or close to, the courting areas of their prey species [29]. The observations reported here indicate that the relocation of *Photuris* females during the hunting period is important to attracting more potential prey (Figure 1). Previous studies (summarized in [12]) indicate that mated females of other predatory firefly species also move from their own courtship display areas to, or close to, the courtship display areas of their prey fireflies; however, the quantitative data (see Appendix A) presented here are the first documenting a possible advantage in female relocation. The location of females within, beside, or between the congregation sites of *Photinus* males—as well as variations in the number of males in these congregations and in the number of responding *Photinus* females and other “competing” *Photuris* females—could result in variations in the number of males interacting with females. The results also provide one of the few published quantitative measures of the importance of firefly aggressive mimicry for attracting prey since *Photuris* females attract *Photinus* males mainly by responding to their flashing signals (Figure 2). However, males apparently are also attracted to other males flashing at *Photuris* females in close range (see [35] for a similar observation). We observed that the glowing responses of *Photuris* females to males of their own species are different from the signals used to attract *Photinus* males (personal observations). It is also interesting that *Photuris* females apparently partially conceal their lanterns by pushing them against the surfaces of their perches as if trying to look smaller, like *Photinus* females. It will be interesting to test this hypothesis. These observations are consistent with the existence of aggressive mimicry in this species. Detailed studies comparing the courtship signals of both species and the hunting signals of *Photuris* are currently underway.

Although *Photuris* femmes fatales attract a considerable number of potential prey at attack distance (Figure 2), their hunting success is low, as only 10% of the females were observed capturing and feeding on a male *Photinus* (Figure 3). These observations are consistent with low success rates documented in most of the few previous studies on attraction and hunting success in *Photuris*, as noted in the Introduction section. Some species of *Photuris* femmes fatales have been observed employing alternative hunting tactics [12,17,30,36,37], and thus, if *P. lugubris* females also exhibit some of these behaviors, it is possible that the low success rate documented in this study could be an underestimation of the actual rates of capture. Two of these strategies involve eavesdropping on the bioluminescent signals produced by their prey: “stalking” and “aerial hawking” [12,17,30,36]. As mentioned in the Results section, in the moments before attacking a deceptively attracted *Photinus*, female *Photuris* turn off their lanterns and attack, a behavior that led one of the reviewers of this paper to suggest that the hunting tactic of *P. lugubris* is a combination of aggressive mimicry and stalking. However, we have never observed a female *Photuris* stalking a prey firefly that she did not attract after it responded to her signals. It can be argued that documenting this behavior requires the continuous observation of perched signaling *Photinus*. During the mating season of 2021, we made observations of female *Photinus* to describe their courtship behavior and never observed a case of stalking by *Photuris*; similar observations of perched males flashing were not made, although this male behavior is relatively rare. With respect to “aerial hawking”, in three consecutive mating seasons (2021–2023), we never observed flying *Photuris* females pursuing and capturing *Photinus* males in flight. However, in 2023, two local tour guides described one observation each of behavior suggesting that this “aerial hawking” strategy might be employed by some females, although probably sporadically. It must be mentioned that the tour guides did not confirm that these observations represented examples of aerial hunting. A third “hunting” strategy reported in *Photuris* fireflies is the stealing of fireflies trapped in spiderwebs (kleptoparasitism) [37]. Although it is common to observe *Photinus* trapped in spiderwebs (but never *Photuris*), frequently with their lanterns on, a case of kleptoparasitism, or even a *Photuris* female close to a spiderweb with *Photinus* trapped, has never been observed. However, with the information available, it is not possible to discard this hunting strategy or consider it rare. Thus, further studies are necessary to determine if female *Photuris* employ other hunting strategies in “Rancho del Valle” and, in case they do, how much prey they obtain in these other ways. What is clear is that many *Photuris* females behave as femmes fatales in our study area and that this strategy has a high rate of failure.

Another intriguing observation of the present study is the fact that all successful hunts occurred during the first attack of the femme fatale, while other females failed after several attacks. This could be the result of individual variation in the ability to deceive and capture male prey by female *Photuris*. Unfortunately, given that female *Photuris* were not individually identified, it is not possible to assess this idea. Studying the possible existence of this variation and its correlates (genetic, developmental, phenotypic, environmental) seems a promising line of research.

One reviewer of this paper pointed out two potential biases in the prey capture estimates. First, the observation that most successful attacks occurred early in the mating period could bias down the prey capture estimate because, later in the night, successful femmes fatales would be hard to detect because they would be feeding instead of interacting with prey. However, we think this possible bias is not very important because *Photuris* females always produce glows, albeit at low rates, when feeding. In fact, thanks to these glows, females already feeding were detected in this study and in a previous one [29]. Second, if there is individual variation in hunting ability (a reasonable expectation), hunting females observed late in the night would tend to be unsuccessful females that would bias down the overall capture rate estimate. This is true; some successful femmes fatales were probably missed early in the night given the limited number of field researchers.

As mentioned in the Introduction, the aggressive mimicry of female *Photuris* is considered a major selective pressure on the communication systems of fireflies in the American continent, and thus, counter-adaptations in prey species are expected to evolve [18]. The low hunting success of *Photuris* femmes fatales found in this and previous studies could be explained by countermeasures that coevolved in the prey [15,18]. Although the existence of a very efficient predatory *Photuris* species cannot be discarded, the general pattern suggests that an asymmetry in the strength of the selective pressures experienced by predators and prey could explain the low hunting success of *Photuris* females and the relatively good ability to escape of the prey fireflies. This asymmetry is known as the “life–dinner principle” [38], which proposes that selective pressures are stronger in the prey species than in the predator because an unsuccessful predator loses a dinner but unsuccessful prey loses its life. In the present study, although *Photinus* males probably paid time, energy (especially when they spent long periods in flight communicating with the predatory females [36]), and opportunity costs by being attracted by *Photuris* females, they were reasonably good at avoiding predation. On the other hand, exploring the possibility of individual variation in *Photinus* males in their ability to avoid being deceived and captured by femmes fatales seems an interesting and complementary line of research for the future.

Although a detailed study of the behavioral strategies used by *P. palaciosi* males to reduce predation by *P. lugubris* females remains to be performed, the “hesitant” and “dropping-when-approached” behaviors observed in males and the “concealing strategies” of females are consistent with the idea of the late Prof. J. Lloyd [18,25] that prey fireflies should evolve a variety of strategies aimed at uncovering the identity of the female they are interacting with. In this context, the absence of “reflex bleeding”, a defense strategy present in several fireflies [39], is intriguing. In three consecutive mating seasons (2021–2023), reflex bleeding was never observed when *Photuris* females attacked males, either in the field or in an experimental setting [29] or when researchers manipulated individuals (both males and females) by hand. In previous studies involving the manipulation of *Photinus* males and females by hand in localities where *Photuris* is absent [31,40] but other predators are present, in the states of Tlaxcala, Puebla, and Estado de México, reflex bleeding was also never observed.

The aggressive mimicry of sexual signals seems to be rare across the tree of life, but the range of organisms exhibiting this behavior is broad. An incomplete list includes female bolas spiders (*Mastophora* sp.) mimicking the sexual pheromones of its moth prey *Spodoptera frugiperda* [5] and males and females of the katydid orthopteran *Chlorobalius leucoviridis* attracting and capturing males of several species of cicadas by responding with sound signals like those of its prey’s females [7]. Pollination via sexual deceit in orchids and other flowering plants is similar because of the imitation of sexual signals; however, successful deception does not mean the death of the cheated receiver [41].

## 5. Conclusions

We measured the attraction of *Photinus palaciosi* males to predatory females of *Photuris lugubris* in the field, as well as the number of prey captured. We observed that females, in general, attract several males of the prey species but capture relatively few (9.8%). Our observations suggest that the low hunting success of *Photuris* femmes fatales could be explained by male behaviors. The attracted males behave in a way that we call “cautious” or “hesitant” that could prevent attacks, and they also behave in ways that probably reduce the success of attacks at a close distance, such as approaching in flight and “dropping” in the face of attack. Females also behave in ways that probably improve their success in attracting and capturing prey, such as relocating in or near the display areas of the prey species, partially concealing the lantern on the substrate, possibly to avoid showing their much larger size and/or to better mimic the responses of *Photinus* females, or turning off their lanterns sometime before attacking. Our observations are consistent with the idea that predators and prey are engaged in a coevolutionary race of adaptation and counter-adaptation. The investigation of possible alternative hunting strategies in *Photuris* and the investigation of individual variation both in the ability to deceive and capture prey by *Photuris* femme fatales and in the ability to detect and escape by *Photinus* males are promising lines for future research.

## Figures and Tables

**Figure 1 insects-15-00078-f001:**
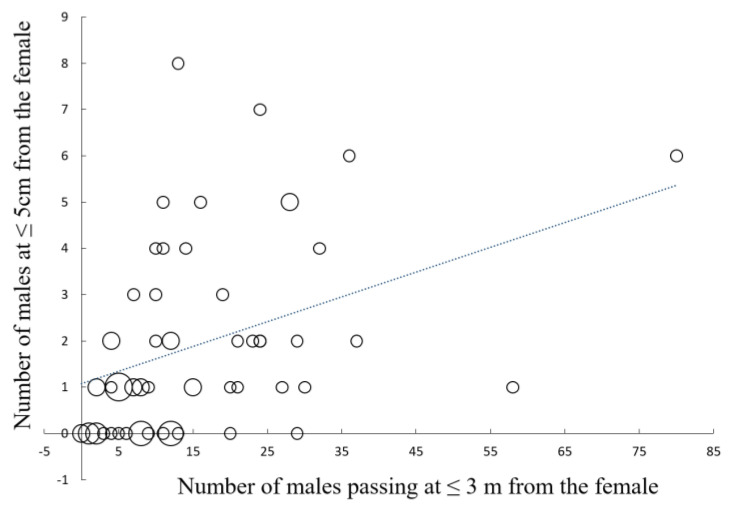
Relationship between the total number of *Photinus palaciosi* males observed flying within 3 m or less of a predatory female *Photuris lugubris* during a nightly courtship period and the number of males approaching that female at 5 cm or less. The size of the circles is proportional to the number of females represented by the circle (1, 2, 3, 4, or 5). The correlation was positive and statistically significant (*r* = 0.46, *p* < 0.00005, *n* = 71).

**Figure 2 insects-15-00078-f002:**
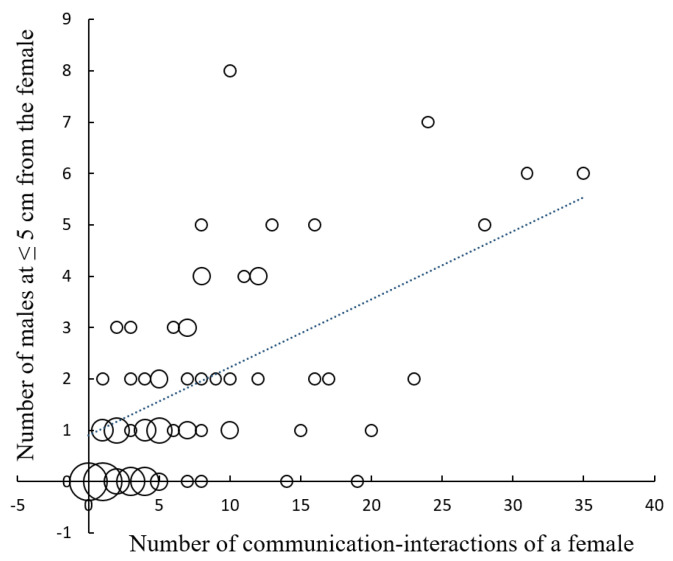
Relationship between the number of “communication” events performed by a predatory female *Photuris lugubris* during a nightly courtship period and the number of *Photinus palaciosi* males approaching that female at 5 cm or less. The size of the circles is proportional to the number of females represented by the circle (1, 2, 3, 4, 5, or 9). The correlation was positive and statistically significant (*r* = 0.65, *p* < 0.00001, *n* = 92).

**Figure 3 insects-15-00078-f003:**
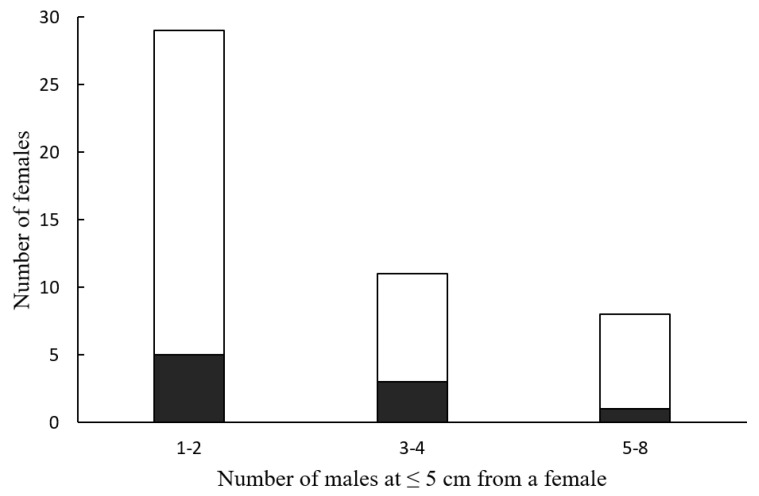
Number of *Photuris lugubris* females that attracted a given number of *Photinus palaciosi* males at 5 cm or less and number of these females that captured one male (shaded area). Fisher’s exact probability test was not significant (*p* = 1).

**Figure 4 insects-15-00078-f004:**
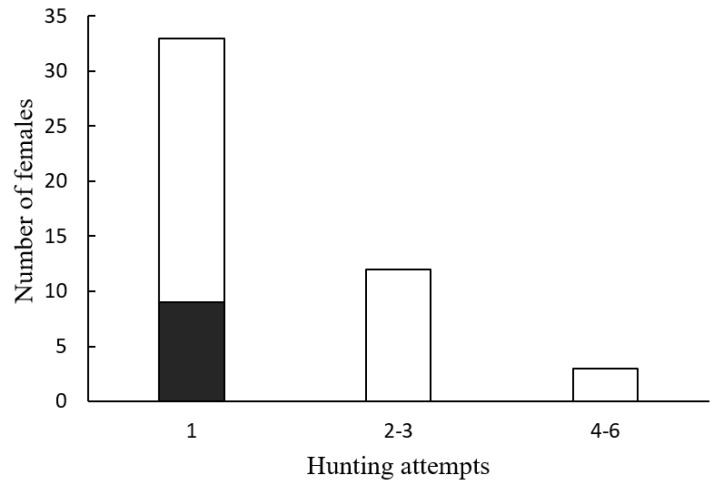
Number of *Photuris lugubris* females that attacked a given number of *Photinus palaciosi* males and number of these females that captured one male (shaded area). Fisher’s exact probability test was significant (*p* = 0.041).

**Figure 5 insects-15-00078-f005:**
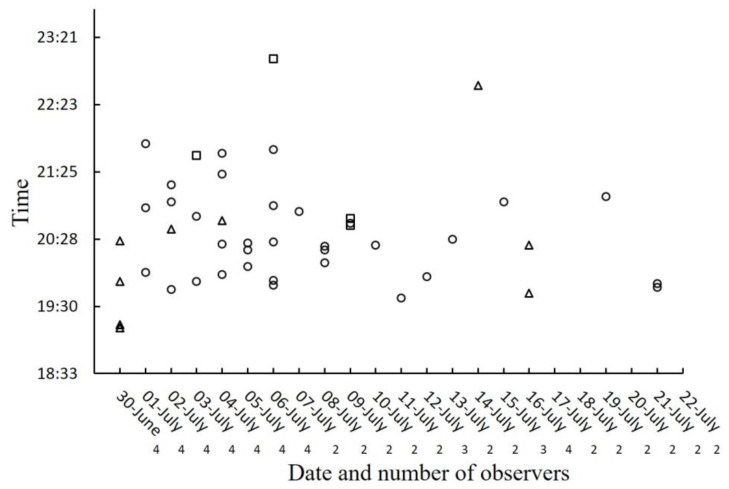
Timing of predator–prey interactions between *Photuris lugubris* females and *Photinus palaciosi* males. Distribution of unsuccessful (the male escaped; circles) and successful (the male was captured; triangles) attacks in relation to the time of the night and date; squares correspond to females that had already captured a male when first observed. The number of researchers in the field is given for each night below the date.

## Data Availability

All the quantitative data analyzed are included in the Appendix A.

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
