# Peer review of "Deceptive Seduction by *Femme Fatale* Fireflies and Its Avoidance by Males of a Synchronous Firefly Species (Coleoptera: Lampyridae)"

_insects, 2024, doi:10.3390/insects15010078_

Round 1

Reviewer 1 Report

Comments and Suggestions for Authors

The paper gives important new information about the success of cheating in the firefly Photuris lugubris females who attract males of Photinus palaciosi for food by using sexual signals of Photinus females. This kind of data are often lacking on “femmes fatales” studies. The authors demonstrate observations following Photuris females in the field.  Although an experimental approach (for example, using fake females) could give more reliable data, this paper is a good starting point for future studies.

I have a few comments to improve the paper:

Figures 1 and 2 should be more informative

Fig.1. What is the biological meaning to count the correlation between number of males approaching far and those approaching close? If the question is of a stepwise process (as the authors claim) the correlation must be positive.

 Instead of correlation I feel that it is more interesting to know how often males escaped after approaching a female from 3 meters distance. The illustration demonstrating that clearly would be more informative. One suggestion is a bar representing mean(+se) number of males approaching female without going closer, a second bar of those approaching both far and close etc. It could give a reader a clear picture on the proportion of males escaping after the remote contact. That information is crucial when estimating co-evolutionary processes on the system.

 Fig 2. Correlation between females signaling and males coming close by. The authors should inform why this picture is important? If the females do not send any signal, how could they attract a male?

Instead of Fig 2. the authors should probably draw a picture with bars where mean (+SE) numbers of communication events of a female that attracted a male close by and those that females that communicated but did not attract a male.

Probably the Fig 1. and also Fig. 2 could be in supplementary material.

Some questions that should be answered to give a better picture of the results:

It was mentioned that:

a) males sometimes “drop” when a female attacked. How often was this observed?

b) Females were noted to hide their abdomen sometimes. How often?

c) Did you ever see Photinus palaciosi females to attract males of its own species and mate? That data could give nice information on the communication differences between cheaters and non-cheaters.

Author Response

The three reviewers of our manuscript (insects-2769147) provided abundant, insightful, and constructive criticism that helped us to improve our manuscript. We are deeply grateful for their thorough revisions. We have followed their suggestions.

“This paper gives important new information…”

A1.1: We thank the reviewer for her/his kind words.

“Figures 1 and 2 should be more informative” “Fig.1. What is the biological meaning to count the correlation between number of males approaching far and those approaching close? If the question is of a stepwise process (as the authors claim) the correlation must be positive.”

A1.2: We have explained more clearly these figures and its importance, both in the Methods (in the statistical analyses paragraph, Lines 243-245) and Results (2nd paragraph, L. 257-263). We also provided the rationale for constructing these graphs in the Introduction (3rd paragraph, L. 95-102).

“Instead of correlation I feel that it is more interesting to know how often males escaped after approaching a female from 3 meters distance. The illustration demonstrating that clearly would be more informative. One suggestion is a bar representing mean (+se) number of males approaching female without going closer, a second bar of those approaching both far and close etc. It could give a reader a clear picture on the proportion of males escaping after the remote contact. That information is crucial when estimating co-evolutionary processes on the system.”

A1.3: We appreciate the suggestions for different ways of presenting the results. However, we feel that Figures 1 and 2 provide a clearer picture of the wide variation observed in the numbers of males: (a) within detection distance of the female (<3m), (b) within attack distance (<5cm) and (c) establishing interactions with the females. We agree with the reviewer that illustrating how often males escaped from the females is crucial, but we think that this information is also provided by figures 3 and 4.

Fig 2. Correlation between females signalling and males coming close by. The authors should inform why this picture is important? If the females do not send any signal, how could they attract a male?

Instead of Fig 2. the authors should probably draw a picture with bars where mean (+SE) numbers of communication events of a female that attracted a male close by and those that females that communicated but did not attract a male.

A1.4: As mentioned above, we expect that the additional explanation of the meaning and importance of this graph in the Methods and Results sections will satisfy the reviewer. As before, we think that Figure 2 provides a clear picture of the variation in the numbers of communication events and its relevance for attracting prey within attack distance. The reviewer is right, female deceptive signalling is crucial for male attraction, however “communication” not only refers to female signalling but to events in which the female actually establishes a “dialogue” and the effect this has on the probability of attracting him within attack distance.

Probably the Fig 1. and also Fig. 2 could be in supplementary material.”

A1.5: For the reasons mentioned above, we decided to maintain these graphs in the manuscript. We hope that the additional explanations mentioned above satisfy the reviewer.

a) males sometimes “drop” when a female attacked. How often was this observed?

  1. b) Females were noted to hide their abdomen sometimes. How often?

A1.6: We did not quantify these behaviours, but in our three years of experience, virtually in all our (numerous) observations of female attacking, escaping males exhibit the “dropping” behaviour. Similarly, all Photuris females observed hunting partially “hid” their lanterns when male Photinus were around. We have made these facts clear in lines 302-307 of the revised version.

c) Did you ever see Photinus palaciosi females to attract males of its own species and mate? That data could give nice information on the communication differences between cheaters and non-cheaters.

A1.7: We have observed many cases. In fact, we have two papers on the mating behaviour of this species (references 31 and 40 of the revised version). We are currently analysing videos of the bioluminescent courtship interactions of P. palaciosi and P. lugubris (as well as of P. lugubris interacting with P. palaciosi males) to describe and comparing the signals of these species. In the revised version, in the first two paragraphs of the new natural history section, we provide a description of the differences in mating behaviour between these two species (L. 138-153 and 160-169). In a previous paper we mentioned that the bioluminescent responses of female Photuris to their own courting males and to their prey are clearly different (though we are still working in a formal, quantitative description of these signals).

Reviewer 2 Report

Comments and Suggestions for Authors

This paper makes an important contribution to our understanding of the aggressive mimicry interactions between Photuris predators and Photinus prey species in North America. As such it is also a valuable addition to the broader literature on aggressive mimicry behavior in general. The authors have conducted a lengthy and substantive set of field observations of firefly behavior over the course of almost a month. From these behaviors they have used correlations to draw important insights into the relationship between Photinus males advertising in the area of a female Photuris and the number approaching to within a few centimeters (a reasonable capture distance), and the relationship between the number of courtship signal communications between a focal Photuris female and any Photinus male and the number of males approaching to within a few centimeters of the focal female. These results clearly illustrate the critical roles of both the density of advertising males and flash signal mimicry for Photuris females to attract Photinus males. While not unexpected, carefully establishing these behavioral interactions is an important contribution as few studies have quantified these interactions in this way. Furthermore, this research establishes the overall rate of capture of Photinus palaciosi by Photuris lugubris, examines the relationship between capturing a male and the number of males within 5cm, and examines the relationship between capturing one male and the number of hunting attempts by a Photuris female. As the authors note, the former is a particularly important contribution as it provides one of the few estimates of the potential selection pressure placed on a prey species by Photuris predation. Finally, the authors provide some observations about the nature of Photuris lugubris and Photinus palaciosi predation and escape behavior.

Unfortunately, both the interpretation of the capture data and of predation and escape behavior represent the weakest elements of an otherwise excellent paper. The analysis of the capture data is fine, but in my opinion this data raises some questions that should be addressed as part of the interpretation within the discussion. Given the finding that there is no evidence of higher capture success with greater number of males attracted, it is important to consider whether other aspects of female predatory behavior apart from aggressive mimicry are potentially better predictors of Photuris hunting success. Photuris have also been observed by Lloyd and other authors, such as Woods et al. 2007 and Lewis et al. 2012 engaging in aerial hawking and stalking behavior (admittedly Lewis et al. 2012 only observed the behavior in captive fireflies). Similarly, given that there appeared to be over 13 females that made 2 or more hunting attempts and never capture a male, while every female that captured a male did so on her first attempt, it is worth considering and discussing whether this reflects intraspecific differences in aggressive mimicry behavior or potentially differences in other aspects of prey capture behavior. The qualitative observations about Photuris lugubris predatory behavior and Photinus palaciosi predator avoidance behavior are valuable contributions and provide a good starting point to expand this discussion. For instance, given that Photuris lugubris appear to partially conceal their lantern and stop responding before attacking, and that Photinus palaciosi appear hesitant to completely approach them, remain flying and even drop when near them, it is easy to imagine that once aggressive mimicry has attracted a male within a reasonable distance, other predatory behaviors that could be better equated with stalking, ultimately determine capture success. Another point relative to the discussion of escape behavior is that while the authors effectively described "hesitant" male behavior in response to aggressive mimicry, in contrasting it to "normal" male courtship interactions they cite a manuscript that at least some of the authors have in preparation. This makes it difficult to assess how different these behaviors truly are, and I wonder whether it would be possible to briefly summarize the "normal" courtship behavior described in that manuscript, so that the distinction is more clear. I believe that doing so will also allow for a stronger discussion about predation and predator avoidance.

One final concern about both the introduction and discussion has to do with the literature review of other papers related to Photuris predation. The authors correctly state that their paper is relatively unique in providing data about predation rates on another firefly species by a Photuris predator. As a result, as I noted above, this paper is a really valuable contribution. However, I think it would be preferable when establishing this in the introduction, rather than directing the reader to the discussion for the review, to provide the relative references in the introduction as well. As far as I can tell, there are only 2 of them, Lloyd 1975 and Lloyd 2018 so it would not be difficult to cite them in the introduction. Furthermore, I believe it would be particularly useful to include one more citation if not in the introduction, then in the discussion. I've mentioned this paper previously, Lewis et al. 2012, which observed predation rates in captivity for a collection of Photuris predators on an interesting range of Photinus, Lucidota, and Phausis fireflies. While the predation rates are not directly comparable because of the artificial nature of observing them in captivity, this data finds a few species, Photinus pyralis, Photinus brimleyi, and Phausis reticulata that have very low predation rates in captivity. This work is also interesting because it also includes one of the only other well-established synchronous firefly species, Photinus carolinus, which in this research was readily consumed by Photuris. This paper raises interesting questions that should be incorporated into the discussion about potential prey defenses beyond those mentioned in this manuscript, such as reflex bleeding and other forms of chemical defense. Additional commentary on these other possible defenses in Photinus palaciosi would strengthen the discussion. As noted above, it might also be useful to include Woods et al. 2007 as a citation to facilitate the discussion of predatory behavior by Photuris. This paper examined Photuris predatory behavior relative to different rates of artificial firefly flashing. Given the authors' observations about Photuris lugubris seeking out areas where Photinus palaciosi courtships occur, it would be useful to cite this study that demonstrated a different Photuris species seeking out artificial Photinus-like flashes. Apart from these concerns about how to discuss the authors findings, I find this paper offers a substantive, important, and well-written contribution to the literature on aggressive mimicry in general and Photuris aggressive mimicry in particular.

I've included the full citations below for the authors' consideration:

Lewis, S.M, L. Faust, and R. De Cock. 2012. The Dark Side of the Light Show: Predators of Fireflies in the Great Smoky Mountains. Psyche: A Journal of Entomology. vol. 2012, Article ID 634027, 7 pages, 2012. https://doi.org/10.1155/2012/634027

Woods, W.A. Jr., H. Hendrickson, J. Mason, and S.M. Lewis. 2007. Energy and Predation Costs of Firefly Courtship Signals. American Naturalist. Vol. 170 (5). pgs 702-708.

vol. 170, no. 5

Comments on the Quality of English Language

Overall the quality of English Language used in this manuscript is excellent. My only suggestions along these lines include replacing men with Homo sapiens when citing reference number 10 and to correct the spelling of lucibufagins in the first paragraph of the introduction. 

Author Response

The three reviewers of our manuscript (insects-2769147) provided abundant, insightful, and constructive criticism that helped us to improve our manuscript. We are deeply grateful for their thorough revisions. We have followed their suggestions.

This paper makes an important contribution to our understanding…

A3.1: We thank the reviewer for her/his kind words. With respect to the initial commentary of this reviewer, we note that he/she agrees with relevance of figures 1 and 2 (“From these behaviors they have used correlations to draw important insights into the relationship between Photinus males advertising in the area of a female Photuris and the number approaching to within a few centimeters (a reasonable capture distance), and the relationship between the number of courtship signal communications between a focal Photuris female and any Photinus male and the number of males approaching to within a few centimeters of the focal female. These results clearly illustrate the critical roles of both the density of advertising males and flash signal mimicry for Photuris females to attract Photinus males. While not unexpected, carefully establishing these behavioral interactions is an important contribution as few studies have quantified these interactions in this way.”).

Unfortunately, both the interpretation of the capture data and of predation and escape behavior represent the weakest elements of an otherwise excellent paper. The analysis of the capture data is fine, but in my opinion this data raises some questions that should be addressed as part of the interpretation within the discussion. Given the finding that there is no evidence of higher capture success with greater number of males attracted, it is important to consider whether other aspects of female predatory behavior apart from aggressive mimicry are potentially better predictors of Photuris hunting success. Photuris have also been observed by Lloyd and other authors, such as Woods et al. 2007 and Lewis et al. 2012 engaging in aerial hawking and stalking behavior (admittedly Lewis et al. 2012 only observed the behavior in captive fireflies).

A3.2: The reviewer is right, in the previous version we failed to consider alternative hunting strategies that have been observed in other Photuris species. In the revised version, we explicitly discuss the alternative strategies mentioned by the reviewer (stalking and aerial hawking), as well as kleptoparasitism in spider webs. We described these alternatives and considered its possible relevance for P. lugubris (briefly, very little evidence of aerial hawking and no observations of stalking and kleptoparasitism, although we cannot discard the last two). We conclude that further studies specially designed to look for these strategies are necessary. However, this lack of information does not affect our conclusions about the success of aggressive mimicry in P. lugubris. This discussion is presented in the second paragraph of the Discussion (L. 358-387).

Similarly, given that there appeared to be over 13 females that made 2 or more hunting attempts and never capture a male, while every female that captured a male did so on her first attempt, it is worth considering and discussing whether this reflects intraspecific differences in aggressive mimicry behavior or potentially differences in other aspects of prey capture behavior.

A3.3: The reviewer is right, in the previous version we failed to consider individual variation in Photuris (or Photinus) behaviour. In the revised version we discuss this possibility. Unfortunately, since we did not mark females, we cannot explore further the data along this line of thought. This discussion is in paragraphs three (L. 390-396) and five (L. 423-426) of the discussion.

The qualitative observations about Photuris lugubris predatory behavior and Photinus palaciosi predator avoidance behavior are valuable contributions and provide a good starting point to expand this discussion. For instance, given that Photuris lugubris appear to partially conceal their lantern and stop responding before attacking, and that Photinus palaciosi appear hesitant to completely approach them, remain flying and even drop when near them, it is easy to imagine that once aggressive mimicry has attracted a male within a reasonable distance, other predatory behaviors that could be better equated with stalking, ultimately determine capture success.

A3.4: This is a good idea: femme fatales probably use a combination of aggressive mimicry and stalking. This could happen always or only when prey seem reluctant to land or approach. We have mentioned this idea in the 4th paragraph of Results (L. 310-313) and in the 2nd paragraph of the Discussion (L. 364-367).

Another point relative to the discussion of escape behavior is that while the authors effectively described "hesitant" male behavior in response to aggressive mimicry, in contrasting it to "normal" male courtship interactions they cite a manuscript that at least some of the authors have in preparation. This makes it difficult to assess how different these behaviors truly are, and I wonder whether it would be possible to briefly summarize the "normal" courtship behavior described in that manuscript, so that the distinction is more clear. I believe that doing so will also allow for a stronger discussion about predation and predator avoidance.

A3.5: All reviewers asked for this information, and we agree is important to include these descriptions. In the revised version, we provide a description of the differences in mating behaviour between these two species in the first (L. 138-153) and second (L. 160-169) paragraphs of the new natural history section. In a previous paper we mentioned that the bioluminescent responses of female Photuris to their own courting males and to their prey are clearly different (reference 29 of the revised version).

Furthermore, I believe it would be particularly useful to include one more citation if not in the introduction, then in the discussion. I've mentioned this paper previously, Lewis et al. 2012, which observed predation rates in captivity for a collection of Photuris predators on an interesting range of Photinus, Lucidota, and Phausis fireflies. While the predation rates are not directly comparable because of the artificial nature of observing them in captivity, this data finds a few species, Photinus pyralis, Photinus brimleyi, and Phausis reticulata that have very low predation rates in captivity. This work is also interesting because it also includes one of the only other well-established synchronous firefly species, Photinus carolinus, which in this research was readily consumed by Photuris. This paper raises interesting questions that should be incorporated into the discussion about potential prey defenses beyond those mentioned in this manuscript, such as reflex bleeding and other forms of chemical defense. Additional commentary on these other possible defenses in Photinus palaciosi would strengthen the discussion.

A3.6: We have incorporated this important reference (also suggested by reviewer 2) in the Introduction (3rd paragraph, L. 106-114) and in the Discussion (2nd paragraph, L. 358-363), when discussion previous studies of hunting success and alternative hunting strategies. Regarding reflex bleeding, a behaviour never observed in P. palaciosi either when attacked by Photuris or when manipulated by hand, we include this behaviour in the discussion (6th paragraph, L. 432-439).

As noted above, it might also be useful to include Woods et al. 2007 as a citation to facilitate the discussion of predatory behavior by Photuris. This paper examined Photuris predatory behavior relative to different rates of artificial firefly flashing. Given the authors' observations about Photuris lugubris seeking out areas where Photinus palaciosi courtships occur, it would be useful to cite this study that demonstrated a different Photuris species seeking out artificial Photinus-like flashes.”

A3.7: Reviewer 2 also suggested this reference. It is indeed relevant and we have included it in the 2nd (L. 358-362) and fifth (L. 420-423) paragraphs of the discussion.

Reviewer 3 Report

Comments and Suggestions for Authors

Author Response

The three reviewers of our manuscript (insects-2769147) provided abundant, insightful, and constructive criticism that helped us to improve our manuscript. We are deeply grateful for their thorough revisions. We have followed their suggestions.

“Thank you for the opportunity to read and review…” “I found this paper to be generally well-written…” Strengths…”

A2.1: We thank the reviewer for her/his kind words.

Weaknesses

The data is available upon request instead of included as a supplementary file

A2.2: We are now including the quantitative raw data as Supplementary Materials.

“Bias in the number of observers across the season may skew results

Potential for bias in how females were sampled over the night may skew results”

A2.3: We consider the two potential sources of bias mentioned by the reviewer (here and later) in the fourth paragraph of the Discussion (L. 396-407). We think one is not very important but the other needs to be assessed.

“Several quotes used in the writing, rather than paraphrasing”

A2.4: We have eliminated all literal quotes but one (the definition of aggressive mimicry).

“Several references to a manuscript in prep (I have not encountered before a manuscript

that has been submitted for review with so many - at least should be loaded onto bioRxiv so that reviewers and readers can see the other manuscript)”

A2.5: We mentioned a “manuscript in preparation” not an “unpublished manuscript”. There is no finished manuscript to submit to arXiv. To avoid confusion, in the revised version we said “personal observations” instead of “manuscript in preparation”.

“Lack of detail on the biology of the predator and prey (e.g. how do they overlap in range? Activity time? Seasonal emergence? Was the sampling period early in the seasonal emergence, late? Does this Photuris species prefer this prey, or other Photinus?)”

A2.6: We now include information about the natural history and behaviour of the species studied. All the questions raised by the reviewer are considered there. This information is in the first three paragraphs of the Materials and Methods section (L. 124-188).

“Major recommendations to the authors:

  1. Include the quantitative data as a supplementary file”

A2.7: We are now including the data as Supplementary Materials (see A2.2).

“2. Add in more discussion of potential biases in the discussion

  1. For example, if an observer stays with the first female that they see, until she flies away or is successful in attack, and most successful attacks occur early in the night, isn’t your estimate of successful attacks potentially an underestimate (you wouldn’t find feeding females later in the night because they are busy eating). And, later in the evening, observers are actually selecting unsuccessful females to observe because successful females are feeding at that point, and thus, not flashing at prey - which would bias the measurement of the successful attack rate down. Just like the male P. palaciosi still signaling at the end of the night – they are the unmated (unsuccessful males).”

A2.8: We consider these potential sources of bias in detail in the in the fourth paragraph of the Discussion (L. 396-407).

3. Consider reading and integrating this source on Photuris predation and feeding preference:

Lewis, Sara & Faust, Lynn & De Cock, Raphaël. (2012). The Dark Side of the Light Show: Predators of Fireflies in the Great Smoky Mountains. Psyche. 2012. 10.1155/2012/634027.”

A2.9: We have incorporated this reference (also suggested by reviewer 3) in the Introduction (3rd paragraph, L. 106-114) and in the Discussion (2nd paragraph, L. 358-363), when discussion previous studies of hunting success and alternative hunting strategies.

4. Consider reading and integrating this source on the evolution of Photuris predation - there does seem to be selection in the predator:

McKinley CN, Lower SE. Comparative Transcriptomics Reveals Gene Families Associated with Predatory Behavior in Photuris femme fatale Fireflies. Genes. 2020; 11(6):627.https://doi.org/10.3390/genes11060627

A2.10: We discuss this reference and two other that also refer to molecular and morphological adaptations of Photuris females to predation in the second paragraph of the Introduction (L. 73-77; see references 19, 20 and 21).

5. Paraphrase, rather than quote, when writing about the work of others (even Jim Lloyd’s most excellent words)”

A2.11: As mentioned above, we have eliminated all literal quotes but one (the definition of aggressive mimicry).

“6. Submit the manuscript in prep to bioRxiv so that reviewers and readers can see for themselves and be convinced by your arguments”

A2.12: We mentioned a “manuscript in preparation” not an “unpublished manuscript”. There is no finished manuscript to submit to arXiv. To avoid confusion, in the revised version we said “personal observations” instead of “manuscript in preparation”.

“7. Add a section of the introduction or methods that describes more about the biology of the predator and prey (and also how the experimental design relates to that)”

A2.13: We include a new section entitled “Firefly natural history in Rancho del Valle”, with information about the basic biology, natural history and behaviour of the species studied. This new section is in the first three paragraphs of the Materials and Methods section (L. 124-188).

“Line 21. I hope that information on how to identify these two species in the field (and whether there are other species active in these areas at the same time) is given in the methods section.”

A2.14: This information is in the new subsection “Firefly natural history in Rancho del Valle” of the Materials and Methods (L. 124-188).

“Line 25. Do these terms “cautious” or “hesitant” anthropomorphize the fireflies too much?”

A2.15: To avoid this problem, these terms are between quotation marks. Besides, at least the term "hesitant" has been used previously in reference to the behaviour of males deceptively attracted by femmes fatales (Lloyd & Wing, 1983).

Line 27-28. It is confusing to just say “females”. You mean Photuris females? Please clarify in

text.”

A2.16: Correction was made.

“Line 28. Would perching be a more accurate term to use than locating?”

A2.17: Suggestion was accepted.

“Line 29. Change to concealing for writing consistency.”

A2.18: Correction was made.

Line 29. The female conceals its own lantern? Or the lantern of the prey? Please clarify in text.”

A2.19: Clarification was made.

Line 30. Unravelling - perhaps revealing or displaying would be more appropriate words to use?”

A2.20: Correction made.

Line 35. North, Central, and South America, correct? Please clarify in text.”

A2.21: Clarification was made.

Line 38. If hunting success is so low (as almost to be random), how does that affect coevolution?”

A2.22: A 10 per cent probability of being captured is not small (it could be a significant selective pressure) and a low hunting rate does not mean hunting is random. In the revised version we provide a broader discussion of coevolution in the context of femme fatale-prey coevolution (see 2nd paragraph of Introduction, L. 77-84).

Line 47. Please clarify in the text - increase reproduction? Food acquisition rather than food

obtaining?”

A2.23: Clarification was made.

Lines 49-50. A quote - please paraphrase.

A2.24: This is the only quote left in the paper. It seems to us that is the more concise definition we can think of. All other quotes were eliminated.

“Line 52. Ha! Thanks for the human reference - I will totally check it out.”

A2.25: You are welcome.

“Line 54-55. What other fireflies do Photuris eat - see Lewis reference above? Please clarify in

text. How might you expect this to affect coevolution?”

A2.26: Since many Photuris eat more than one prey species, in the revised version we include a discussion of the consequences of this fact for coevolution (2nd paragraph of Introduction, L. 77-84).

“Line 64. Photuris species have been shown to provide the appropriate response….? Please clarify in the text.”

A2.27: Clarification was made.

Line 65. How large is the flash repertoire of a Photuris? Are they generalists or specialists? How

is this expected to affect coevolution - usually I’d expect specialists to coevolve rather tan generalists? Please clarify in text.”

A2.28: Clarification was made in the Introduction (2nd paragraph of Introduction, L. 77-84).

“Line 67. Another quote - please paraphrase.”

A2.29: Correction was made.

Line 69 -72. A long and confusing sentence. Can you please rewrite it to make it less confusing - I think it is an important sentence in this paper.”

A2.30: Modification was made (see 2nd paragraph of Introduction, L. 77-80).

Line 79 - 81. Another quote - please paraphrase.” “Line 88 - 90. Another quote - please paraphrase. This quote is also confusing to read with added firefly-specific words.”

A2.31: Corrections were made.

Line 96. Does a prey need to be actively hunted to count as “Escape”. How does fit in the experimental design - is it # communicated - # captured?”

A2.32: We decided to eliminate this sentence as it describes an obvious fact.

Line 103. How does this sampling period relate to the beginning and end of the season emergence of these two firefly species? How much do their seasonal emergences overlap? Are they seen in the same hábitat always, or is one more limited in the habitat type than the other? What times of night are each active, how does that correspond to the observation period? Are Photuris that are signaling likely mated, virgin, ovipositing, not ovipositing? How is Photuris hunting expected to change over the season? Suggest reading/integrating this reference:

Zorn, L.P.; Carlson, A.D. Effect of mating on response of female Photuris firefly. Anim. Behav. 1978, 26, 843–847”

A2.33: Most of the information asked is included in the “Firefly natural history…” subsection. With respect to the relation between mating and hunting in Photuris, we discussed this in more detail also in the natural history section (L. 142-153) and include the suggested reference and another on the same subject (references 32 and 33).

Line 111. Flew away? Stop performing? Just checking grammatical tense.”

A2.34: These grammatical corrections were made.

Line 112. Stopped responding to male flashes - you mean the female was tested? Or didn’t communicate with any prey males? Did you ever observe a female doing a mating flash instead? Will the femme fatales eat their own males?”

A2.35: When the female stopped responding to prey, we usually checked if she was present and wait until the courtship period of Photinus finished; we did not tested the female. The overlap between mating and hunting is discussed in the natural history section (L. 142-153). Yes, some hunting females are still sexually receptive. In a previous study (Maquitico et al, 2022), we tested female Photuris for multiple mating and left male-female pairs in plastic containers during the whole night (N = 14 females confined with between 4 and seven males, one per night). In two cases, the female killed and partially ate the male (only part of the thorax contents). There are a few other published reports of cannibalism in captivity and are believed to be laboratory artefacts (males and females apparently never remain together after mating). We never observed cannibalism in the field.

Line 120. Another quote - please paraphrase.

A2.36: Correction was made.

“Line 123. How could observers tell that the perched female was responding to a particular flash and not just flashing at random?”

A2.37: This is now explained in the Methods section (2nd paragraph of the Field Study subsection, L. 216-223).

“General methods comment. What about hawking behavior? Did you observe a femme fatale hawking a prey male? Does this result in an underestimate of successful attacks?”

A2.38: In the revised version, this and other alternative hunting strategies are considered in the second paragraph of the Discussion (L. 357-388). Briefly, we have never observed aerial hawking, but two tourist guides described two observations suggesting that this behaviour could occasionally be employed.

“Line 133 - 135. Presumably it is dark out when this happens. If the female lantern is off, how were observations of successful attacks made?”

A2.39: Sometimes moonlight allowed observing attacks, in other cases we stopped observing the flashing male and checked for his presence (usually with the dim light produced by the screen of the cellphone or with a small lantern producing a dim blue light), in other cases captured males produce continuous light (similar to the "distress light produced by fireflies captured in spider webs) and, finally, Photuris females produce glows at low rate when feeding.

Line 136. Did you observe prey, in unsuccessful attacks, reflex bleeding? It is hypothesized as a defensive strategy that could support your coevolution argument.”

A2.40: There is no evidence of reflex bleeding in Photinus. In the revised version we discuss this in the sixth paragraph of the Discussion (L. 431-438).

Line 140. Pearson correlation assume linearity of your data. Did you test for this?”

A2.41: No, but most statistical texts suggest its use when sample sizes are relatively large (> 30). On the other hand, by using non-parametric correlations we obtained the same results.

Line 158 -159. I think there is an error in the writing here. Females passing at a distance?

Please clarify in the text.”

A2.42: Thank you for spotting this error. Correction was made.

Lines 167. It looks like there are a lot of Photuris females with zeros. Doesn’t that mean they are not hunting? Shouldn’t they be eliminated from the analysis?”

A2.43: All females included in the analyses were hunting (they were away from their initial courting area and they were responding to male Photinus flashes), thus excluding them would artificially increase estimates of attraction and hunting success. Besides, the number of females located in places where no males were observed is very small. Instead, we observed several females that did not succeed in attracting males within attack distance.

Line 183. Do you think it could also be that female Photuris with previous experience may be the ones to be first to catch a prey, and thus, catch earlier in the night? How many prey will a Photuris eat…in a night? In a season?”

A2.44: It is a reasonable hypothesis. A female takes several hours to feed on one male, thus she cannot feed on more than one per night. As mentioned in the new Natural History subsection, we observed one female discard a partially eaten prey to mate. If this female (or a female in this situation) captures a second prey after mating, she could kill two male Photinus but only would feed on one and part of the other. How many in a season, we have no idea.

“Line 182 - 185. This sentence is confusing. Please re-write.”

A2.45: Correction was made. We deleted the commentary about the time taken to consume one prey because we mentioned this in the Methods section (2nd paragraph of the Field Study subsection, L. 233-234).

“Line 185. Should there be a citation for taking several hours to eat a Photinus?”

A2.46: We observed this in a previous experiment on feeding in captivity (Maquitico et al. 2022), but we did not report it. Our field observations are also consistent with long feeding times, but since we have not made continuous observations of females feeding we cannot be sure. We can only say that when we retourn the same night, one or two hours after they were first observed feeding, they are still feeding.

Line 197. How does “hesitant” behavior differ from “normal” behavior? What is normal?”

A2.47: We explained “normal” (intraspecific) behaviour in the Natural History subsection. Then we explained “hesitant” behaviour in the fourth paragraph of Results (L. 295-302).

Line 201. Suggest putting manuscript on bioRxiv, so can cite in this submission and establish precedence.” “Line 237. BioRxiv?” “Line 242. BioRxiv?”

A2.48: We do not mention an "unpublished manuscript" but a "manuscript in preparation". So, there is no finished manuscript to upload in bioRxiv. We have changed "manuscript in preparation" for "personal observations" to avoid confusión.

Line 203. Drops off leaf? In the air? Do Photuris females follow?”

A2.49: This is now explained in the fourth paragraph of Results (L. 302-305).

Line 204. Why is partially italicized?”

A2.50: It was a mistake.

Line 205. Do you think they are trying to adjust the lantern size to better match the lantern of the prey female, or their whole body size?”

A2.51: We do not know.

Line 209. SUggest switching other and four to read “four other” for readability.”

A2.52: Suggestion accepted.

Line 214. Do you think the decrease is related to more females moving to oviposition?”

A2.53: Possibly, but we do not know.

Line 216. Again - what is the overlap with activity time and observation?

A2.54: This is explained in the Natural History subsection.

“Figure 5. Shouldn’t prey males be expected to be less discriminating (more desperate) at the end of the night/end of the season and thus, more susceptible to deception?”

A2.55: It is possible, but there are many unknowns. We do not know if these are late-emerging males or "old" males, because we do not know the pattern of emergence (although, obviously, lots of males emerge at the beginning of the mating season). We do not know if these males mated previously or if they are still virgin.

Figure 5. The first unsuccessful and successful attacks are observed at 18:57- isn’t that before dark/flashing has stated?”

A2.56: In the new Firefly Natural History subsection, we explained that most activity starts at 19:30, but some individuals can be observed as early as 19:00 h. Besides, there were two mistakes in this graph and we are very sorry: the two captures were observed at 19:12 and 19:15, and there was another misplaced point at 18:45 of a escaped male that it really occurred at 20:52. Of course, after spotting these errors, we checked all numbers in the data base, graphs and analyses. Fortunately, these were the only mistaken values.

Figure 5 - definitely seems like fewer observers = fewer observations. Glad you mentioned this in the paper.”

A2.57: Thank you.

Line 225. So female Photuris place themselves close to congregating prey males in order to attract more prey? I think the wording of this sentence is confusing me.”

A2.58: We modified the wording (first paragraph of Discussion, L. 329-331).

Line 229. How large are congregation sites? How close together?”

A2.59: This varies a lot within the mating season. Sometimes from one nigth to the next.

“Line 239. Can you stress that this is hypothesized? I think the work still needs to be done to see if the Photuris is actually doing this and if it has any effect.”

A2.60: The reviewer is right and we have followed his/her advice (first paragraph of Discussion, L. 349).

Line 263. Prey.” “Line 265. Life-dinner” “Line 267. Dinner

A2.61: All these corrections were made.

Line 268. Flight is actually the most expensive thing. Suggest read and incorporate this reference:

Woods, William & Hendrickson, Holly & Mason, Jennifer & Lewis, Sara. (2007). Energy and Predation Costs of Firefly Courtship Signals. The American naturalist. 170. 702-8. 10.1086/521964.

A2.62: We have cited this relevant paper in the fifth paragraph of the Discussion (L. 419-422).

LIne 273 - 278. Another quote. Please paraphrase.

A2.63: Correction was made.

Line 279. Rare across the tree of life, instead of uncommon?

A2.64: Suggestion accepted.

Lines 279-286. This paragraph seems a bit out of place - it zooms out to look across taxa, but then the conclusion zooms back into the present study and fireflies. Perhaps this should be a second concluding paragraph?

A2.65: We understand the point, but decided it is important to compare with other cases of aggressive mimicry.

Line 297. Conceal changed to concealing. Unravelling changed to displaying?” “Line 300. prey

A2.66: Suggestions accepted.